# Neurofilament Light Chain as A Biomarker for Brain Metastases

**DOI:** 10.3390/cancers12102852

**Published:** 2020-10-02

**Authors:** Anne Winther-Larsen, Claus Vinter Bødker Hviid, Peter Meldgaard, Boe Sandahl Sorensen, Birgitte Sandfeld-Paulsen

**Affiliations:** 1Department of Clinical Biochemistry, Aarhus University Hospital, 8200 Aarhus N, Denmark; Claus.Hviid@auh.rm.dk (C.V.B.H.); boesoere@rm.dk (B.S.S.); Birgitte.Sandfeld@aarhus.rm.dk (B.S.-P.); 2Department of Clinical Biochemistry, Viborg Regional Hospital, 8800 Viborg, Denmark; 3Department of Oncology, Aarhus University Hospital, 8200 Aarhus N, Denmark; petemeld@rm.dk

**Keywords:** brain metastasis, neurofilament proteins, biomarkers, cancer, diagnostic

## Abstract

**Simple Summary:**

Early detection of brain metastases is warranted to allow timely intervention that can improve local control and survival time. Neurofilament light chain (NfL) is a neuron-specific protein released after neuronal decay, and we evaluated blood-borne NfL as a biomarker in 43 lung cancer patients with brain metastases and 25 without brain metastasis. NfL was elevated in patients with brain metastasis and serial measurements uncovered increasing NfL levels with a median of three months before a brain metastasis was found on a brain scan. Our findings imply that measuring chances of NfL in the blood could be a potential biomarker for early detection of brain metastases.

**Abstract:**

Background: Brain metastases are feared complications in cancer. Treatment by neurosurgical resection and stereotactic radiosurgery are only available when metastatic lesions are limited and early detection is warranted. The neurofilament light chain (NfL) is a sensitive neuron-specific biomarker released following neuronal decay. We explored serum NfL as a biomarker of brain metastases. Methods: Serum was collected from 43 stage IV lung cancer patients with brain metastases and 25 stage I lung cancer patients. Serum was collected at time of cancer diagnosis and at time of brain metastasis diagnosis. In nine patients with brain metastases, additional samples were available between the two time points. NfL was quantified by Single Molecule Array (Simoa)™. Results: The median NfL level was significantly higher in patients with brain metastases than in patients without (35 versus 16 pg/mL, *p* = 0.001) and separated patients with an area under the curve of 0.77 (0.66–0.89). An increase in NfL could be measured median 3 months (range: 1–5) before the brain metastasis diagnosis. Further, a high level of NfL at time of brain metastasis diagnosis correlated with an inferior survival (hazard ratio: 2.10 (95% confidence interval: 1.11–3.98)). Conclusions: This study implies that NfL could be a potential biomarker of brain metastases.

## 1. Introduction

Brain metastases are devastating and feared complications in cancer patients as they lead to decreased quality of life and reduced patient survival [1,2,3]. The metastases are commonly treated with multimodal salvage therapies including a combination of surgery, radiotherapy, chemotherapy or targeted therapies. Among these, neurosurgical resection or stereotactic radiosurgery are superior to improve local control and survival but can only be provided to patients with a limited metastatic burden [4]. It is therefore of utmost importance to diagnose brain metastases as early as possible. Current routine diagnosis is done by computed tomography (CT) or magnetic resonance imaging (MRI) but is rarely performed before clinical symptoms develop. Consequently, around two-thirds of the patients have multiple metastases at the time of diagnosis [4] and cannot be offered directed therapy. The identification of alternative biomarkers to aid the diagnosis of cerebral metastasis early along the course of the disease is therefore warranted.

Neurofilaments comprise a group of neuron-specific cytoskeletal proteins that are abundant in neurons and confined to the intracellular environment [5]. They are released to the extracellular space following axonal damage and previous studies have demonstrated solid biomarker capabilities of neurofilaments in acute as well as chronic central nervous system (CNS) diseases [6,7,8,9,10,11]. Early studies determined neurofilament levels in the cerebrospinal fluid (CSF) but the recent development of ultra-sensitive technologies such as the Single Molecule Array (Simoa) now allow their detection also in the peripheral circulation [12]. The collective literature demonstrates a close correlation and similar biomarker capabilities of CSF and blood-based neurofilament measurement [6], which opens the venue for the use of serum neurofilament levels for routine biomarker purposes. Neurofilaments levels are highly sensitive to even minor injuries such as concussions [9], which indicate a potential to uncover brain metastatic lesions early in the process.

In this study, we firstly explored if serum neurofilament light chain (NfL) levels are elevated in lung cancer patients with brain metastases compared with lung cancer patients without brain metastases. Secondly, we investigated if the occurrence of brain metastasis was associated with elevation in serum NfL levels and whether such elevation could be detected before the clinical diagnosis of the metastasis. Lastly, we evaluated if the NfL level had a prognostic value in patients with brain metastasis

## 2. Results

### 2.1. Patients

In this retrospective cohort study, a total of 43 stage IV non-small cell lung cancer (NSCLC) patients with brain metastasis were included. Baseline characteristics of the patients are described in Table 1. The majority of patients had adenocarcinoma and were former or current smokers. Twenty-seven of the 43 stage IV patients had their brain metastasis at time of lung cancer diagnosis, while the brain metastasis developed median 11 months (range: 2–19 months) after the lung cancer diagnosis in the remaining 16 patients.

### 2.2. NfL as A Diagnostic Biomarker

At time of lung cancer diagnosis, the median NfL level was comparable between the stage I patients and the 16 stage IV patients without brain metastasis at diagnosis 16 pg/mL (range: 8–165; interquartile range (IQR): 13–25) vs. 20 pg/mL (range: 6–54; IQR: 13–32; *p* = 0.702, Figure 1), indicating that stage of disease has no influence on the NfL level. Yet, in the 27 patients with brain metastasis at time of lung cancer diagnosis, the NfL levels were significantly elevated (median 34 pg/mL (range: 11–151; IQR: 20–65)) compared with stage IV patients without brain metastasis (*p* = 0.015, Figure 1). By using the NfL level measured at time of brain metastasis diagnosis from all 43 stage IV patients, a separation from the stage I patients could be performed with an area under the curve (AUC) of 0.77 (95% confidence interval (CI): 0.66–0.89, *p* = 0.0002, Figure 2A). In this dataset, the optimal cut point was 24 pg/mL, giving a sensitivity of 69% (95% CI: 52–83%), a specificity of 76% (95% CI: 55–91%), a positive predictive value of 79% (95% CI: 62–91%) and a negative predictive value of 60% (95% CI: 41–77%).

### 2.3. NfL as A Predictive Marker

The NfL level was measured both at the time of lung cancer diagnosis and at the time of brain metastasis diagnosis in 13 patients of the 16 patients who developed brain metastasis after the lung cancer diagnosis (Figure 2B). In 12 out of these 13 patients, the NfL level increased between the two time points, whereas a decrease was observed in one patient only. The median percentage change was 289% (range: –14–3790%).

Nine patients had multiple sample points from the time of lung cancer diagnoses to the time of brain metastasis diagnosis. The trajectory for the individual patient is illustrated in Figure 3. In all patients, NfL levels were increased at the time of brain metastasis diagnosis compared with the time of lung cancer diagnosis. Interestingly, increased NfL could be measured with a median lead time of 3 months (range: 1–5 months) from the first increased NfL level to the time of brain metastasis diagnosis. Transient peaks in NfL level not related in time to the brain metastasis diagnosis were found in four patients. In three of these (patient ID 1492, 1682 and 1728), the transient increases were significant (median 480% (range: 352–1119%)), and all correlated in time with the patient receiving chemotherapy and thoracic radiation against their lung tumor.

### 2.4. NfL Level as A Prognostic Marker

The overall survival (OS) of all patients with brain metastases was 5.4 months (95% CI: 0.8–36.1 months). Dividing the patients into a low and high NfL level group according to the median NfL level, a significant longer OS was found in the low level group (10.1 months (95% CI: 2.6–36.1 months)) compared with the high level group (4.1 months (95% CI: 0.8–21.7 months), *p* = 0.01, crude hazard ratios (HR): 2.10 (95% CI: 1.11–3.98), Figure 4). To account for differences in treatment modalities between the patients, the survival analysis was repeated only in the group of treatment-naïve patients, excluding patients harboring epidermal growth factor receptor mutations (*N* = 24). This confirmed the association between high NfL (*N* = 12) and inferior survival (crude HR: 2.43 (95% CI: 1.03–5.69)).

## 3. Discussion

In this study, we evaluated NfL as a potential biomarker for brain metastases in patients with NSCLC. We observed significantly increased NfL levels at time of brain metastasis diagnosis compared with time of lung cancer diagnosis, and NfL measurements provided a fair diagnostic separation of patients with brain metastases and patients without brain metastases. Most interestingly, an elevated NfL level could be measured in serum prior to the patient being diagnosed with the brain metastasis in all except one patient. Lastly, the level of NfL at time of brain metastasis diagnosis was found to correlate with survival. 

Our observation of elevated levels of NfL at time of brain metastasis diagnosis is in line with the collective literature on NfL which have demonstrated elevated NfL levels in diseases associated with neuronal decay [6,7,8,10,13]. While the specific nature of NfL liberation after neuronal damage is unknown, elevations have been observed in a variety of mechanisms from trauma [9,13], degenerative processes [7,10,14], ischemia/anoxia [11,15], human immunodeficiency viruses-infections [16] and hematoma compression [17]. Hence, a release of NfL would be expected as a consequence of the neuronal damage caused by the infiltration of the brain metastasis in the brain parenchyma [18] and/or brain compression caused by the metastasis. In line with this, a recent study found elevated NfL levels in seven cancer patients with brain metastasis compared with healthy controls [19], which supports our findings.

Yet, the most interesting observation in our study was the predictive value of the NfL dynamic in patients. By monitoring the level in consecutive samples, we found that an increase in NfL level could be measured in all except one patient prior to the development of symptoms leading to a brain scan, and that the increase could be observed median 3 months before the symptoms. This observation is consistent with a previous investigation demonstrating that even minimal brain injury could be detected by NfL measurements [9,20,21]. Accordingly, athletes exposed to repeated minimal head impact experienced increasing NfL levels without neuroradiological evidence of nerve damage [9,21]. This suggests that NfL is a highly sensitive biomarker to detect nerve affection and that an increase in NfL level could be an important early indicator of a brain metastasis, indicating that a brain scan should be performed. In addition, a recent study supports the predictive value of the NfL dynamic within patients as longitudinal measurements of NfL were found to hold more prognostic information on long-term outcome than a single NfL measurement in multiple sclerosis patients [22].

In three patients, a transient peak in NfL level was observed in concordance with the patient receiving chemotherapy and radiation against their lung tumor. NfL is expressed also in peripheral fibers, and it has previously been found that damage of periphery nerve cells can lead to elevated NfL level [23]. Consequently, the observed increased NfL level in our patients may be explained by periphery nerve cell damage caused by radiation and/or chemotherapy-induced peripheral neuropathies. Thus, further research is needed to clarify the treatments influence on the NfL level.

Lastly, we observed that NfL level could hold a prognostic value in patients with brain metastases as a significant reduced survival was found in patients with high levels. Estimation of a patient’s prognosis is essential for guidance of clinicians in the optimal management of the patient’s treatment and NfL measurements could potentially add further value to the assessment performed by clinicians, leading to a more accurate estimations of the patient’s lifespan.

This study is the first to qualify NfL as a potentially biomarker of brain metastasis in cancer patients. Yet, the study has some limitations to consider. The number of patients was limited, and our results are primarily hypothesis-generating. Yet, despite the paucity of data, we found indications of very interesting biomarker capabilities. Moreover, consecutive blood samples were not collected with the same timespan before the diagnosis of the brain metastasis in patients, and neuroimaging was performed when the patient developed symptoms indicating brain metastasis and not consecutively at fixed time points. Thus, no scan was performed in accordance with the observed NfL increase, and we cannot determine if a NfL increase actually could be measured prior to the metastasis being visual on neuroimaging. Nevertheless, the NfL increase preceded the development of symptoms. Further, we did not have information on the intrathecal tumor burden, which could in fact be closely associated with the NfL level. It is plausible that patients with minimal intrathecal tumor burden have NfL levels within the normal range. Finally, based on the design of the study, we only evaluated the NfL dynamic in selected patients with confirmed brain metastases. Thus, we are unable to exclude that a change in NfL over time could be found independent of the development of brain metastasis. Therefore, it would be important in the future to investigate if a dynamic in NfL exists in stage IV patients without brain metastases.

## 4. Materials and Methods

### 4.1. Patients

Patients referred to the Department of Pulmonary Medicine, Aarhus University Hospital, Denmark, under the suspicion of having lung cancer were recruited from April 2011 until September 2014. The study population has previously been described in detail [24] and has been used in other publications. In short, 1735 patients were included in the study population, of whom 430 were diagnosed with lung cancer. A baseline blood sample was collected from all patients at time of inclusion and samples were drawn from patients with confirmed lung cancer upon visit to the outpatient clinic at the Department of Oncology, Aarhus University Hospital, Denmark. All patients were treated according to the standard protocol in our institution. Treatment response was evaluated on CT scans and defined according to Response Evaluation Criteria in Solid Tumors version 1.1 [25]. Neuroimaging was performed if the patient had clinical symptoms that gave suspicion of brain metastases. During treatment periods, patients were seen in the outpatient clinic every 3 weeks, while a visit was planned every 12 weeks between treatments. 

For this study, patients with pathological confirmed NSCLC and CT- or MRI-verified brain metastases (stage IV disease) were included if a blood sample drawn within 14 days from the date of the brain metastasis diagnosis was available. As the NfL level has not previously been evaluated in lung cancer patients, we further included a group of consecutive stage I NSCLC patients with a minimum follow-up time of five years without a brain metastasis to determine the NfL level in lung cancer patients. The risk of asymptomatic brain metastasis at time of diagnosis in these patients was negligible. Clinicopathological characteristics were collected at time of inclusion in all patients. The study was conducted according to the Helsinki Declaration and all patients gave informed written consent before inclusion. The study was approved by the Central Denmark Region Committees on Biomedical Research Ethics (no. 1-10-72-39-19) and the Danish Data Protection Agency (file no. 1-16-02-346-14).

### 4.2. Blood Samples

Ten mL of blood was collected into serum tubes (BD vacutainer^®^, Franklin Lakes, NJ, USA) at each time point. The samples were allowed to clot for 30 min before being centrifuged for 15 min, 1400× *g* at room temperature (22–24 °C). Subsequently, serum was isolated and frozen at –80 °C until further analysis. In this study, NfL level was quantified in all available blood samples collected from time of lung cancer diagnosis to time of brain metastasis diagnosis in all stage IV patients in the cohort. In the stage I patients, NfL level was only measured in the blood sample collected at time of lung cancer diagnosis.

### 4.3. Laboratory Analysis

The quantification of NfL levels has previously been described [26]. In short, the NF-light^®^ assay was established on the ultra-sensitive Simoa™ HD-1 platform (Quanterix^©^, Lexington, MA, USA) [12]. According to the manufacturer, the limit of detection and limit of quantification for NfL are 0.038 and 0.174 pg/mL, respectively. The calibrator range is 0–500 pg/mL with linearity from 4–128 times dilution. In our laboratory, the intra- and inter-assay coefficients of variation are 4.3% and 6.4%, respectively. Positive controls in two levels (3.63–5.71 and 125–187 pg/mL) were supplied with the NF-light^®^ assay and included in duplicates in each run. A volume of 68 µL patient serum was used for each analysis. 

### 4.4. Statistical Analysis

A power calculation was performed by Sattertwait’s test on the endpoint of elevated NfL levels among patients with brain metastases compared to patients without brain metastases. Applying a two-sided alpha level of 0.05 with the included study group, the power to detect a two-fold increase among patients was 98%. Baseline characteristics were compared between patients with brain metastases and patients without brain metastases using the Mann–Whitney rank sum test. Data distribution on NfL levels were presented by median (range) and compared by the Mann–Whitney rank sum. Paired observations were compared by the Wilcoxon matched pairs signed rank test. The diagnostic value was estimated by AUC and receiving operator curve (ROC). Last follow-up date was 21 December 2018. OS was determined as the time from the day of the diagnosis of brain metastasis until death of any course or last follow-up date. Estimates of median OS were calculated using the Kaplan–Meier method and analyzed by the log-rank test. Univariate HR were determined using the Cox proportional hazards model. All tests were two-sided and a *p*-value < 0.05 was considered statistically significant. Analyses were performed using the statistical software GraphPad PRISM^©^ and STATA version 15 (StataCorp LP, College Station, TX, USA).

## 5. Conclusions

Altogether, our study indicated that longitudinal serum NfL measurements could be valuable to monitor NSCLC patients for development of brain metastases and thus serve as an important predictive biomarker. The study also suggested that serum NfL may have prognostic potential in NSCLC patients who already have developed brain metastases and calls for further investigations to illuminate this. If these data are validated, NfL could in the future be used as a non-invasive method for surveillance of asymptomatic patients and hopefully lead to an earlier detection of brain metastases. The study calls for further investigations into the potential biomarker capacity of NfL in cancer patients.

## Figures and Tables

**Figure 1 cancers-12-02852-f001:**
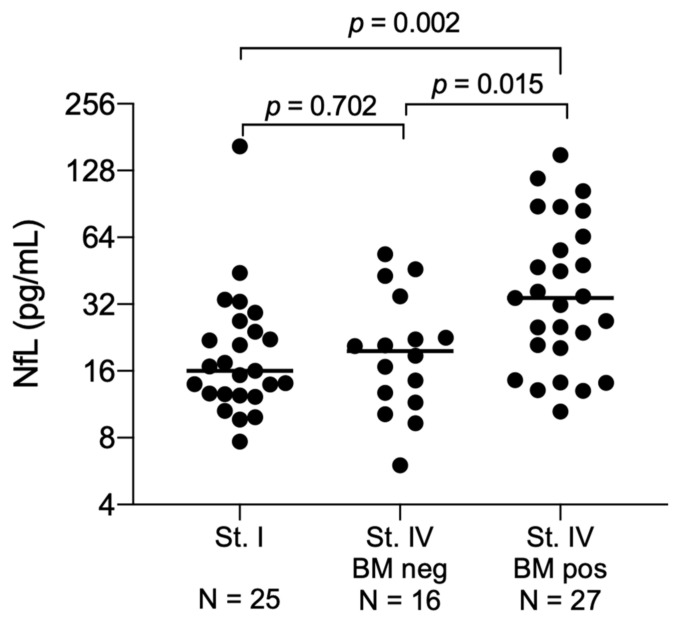
Neurofilament light chain levels at the time of lung cancer diagnosis. Patients are divided in stage I patients (*N* = 25), stage IV patients without a brain metastasis at time of lung cancer diagnosis (*N* = 16) and stage IV patients with a brain metastasis at time of lung cancer (*N* = 27). Median values are marked for each group. Medians are compared by the Mann–Whitney rank sum. Abbreviation: NfL: Neurofilament light chain; St: Stage; BM; Brain metastasis; neg, negative; pos, positive.

**Figure 2 cancers-12-02852-f002:**
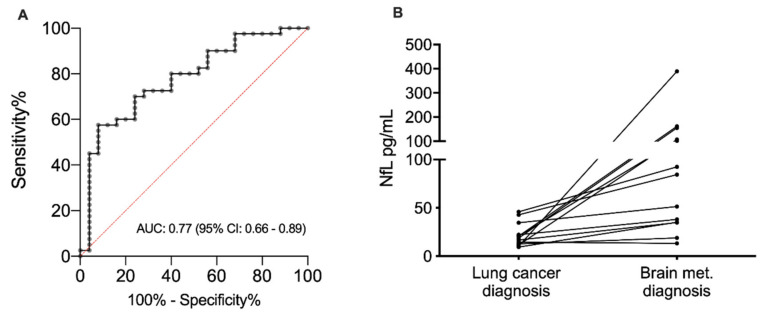
(**A**) Receiving operator curve showing the diagnostic value of the neurofilament light chain level separating non-small cell lung cancer patients with brain metastasis (*N* = 43) and stage I non-small cell lung cancer patients (*N* = 25). The diagnostic value is estimated by the area under the curve and presented with a 95% confidence interval. (**B**) Change in neurofilament light chain level from the time of lung cancer diagnosis to the time of brain metastasis diagnosis in lung cancer patients with available blood samples at both time points (*N* = 13). Abbreviation: AUC, area under the curve; CI, confidence interval; NfL: Neurofilament light chain; met: metastasis.

**Figure 3 cancers-12-02852-f003:**
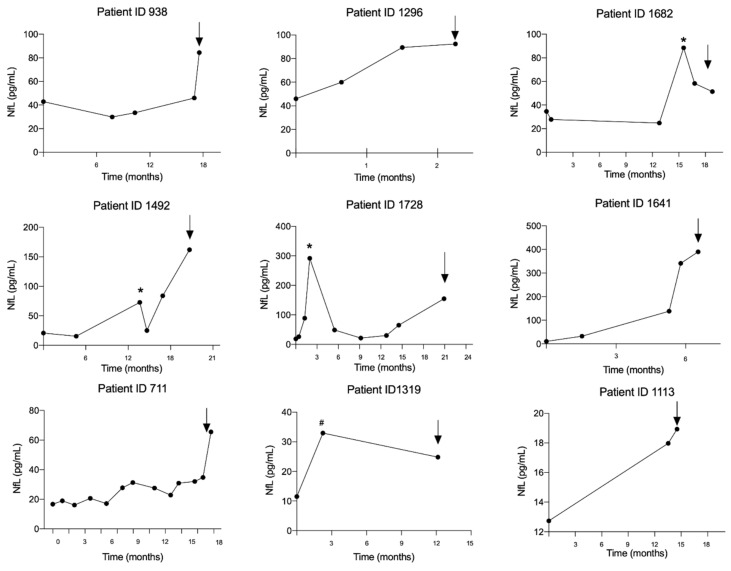
The trajectories of the neurofilament light chain level from time of lung cancer diagnosis to the time of brain metastasis in nine lung cancer patients. The arrows indicate time of brain metastasis diagnosis. The asterisks (*) marks the time point where the patient has received both chemotherapy and thoracic radiation against their lung tumor and the hashtag (#) marks the time point where the patient has received only chemotherapy. Abbreviation: NfL: Neurofilament light chain.

**Figure 4 cancers-12-02852-f004:**
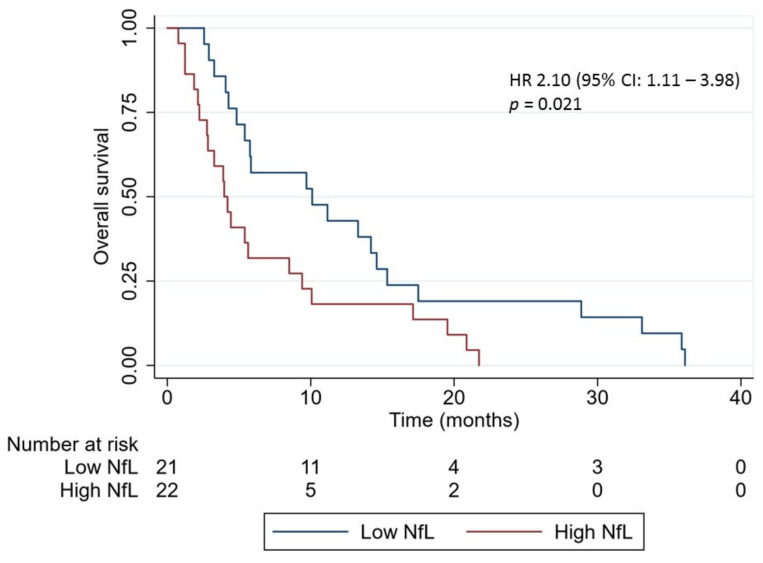
Kaplan–Meier curves for overall survival in lung cancer patients with brain metastasis according to the neurofilament light chain (NfL) level at time of brain metastasis diagnosis (*N* = 43). Patients were divided in a low and high NfL level group according to the median NfL level. *p*-value was calculated by the log-rank test. Abbreviation: NfL: Neurofilament light chain; HR: hazard ratio; CI, confidence interval.

**Table 1 cancers-12-02852-t001:** Patient characteristics.

Characteristics	Patients, *N* = 43
Male, *N* (%)	22 (51)
Age, Median (range)	64 (41–85)
Histology, *N* (%)	
—Adeno	38 (88)
—Squamous cell	5 (12)
Brain metastasis present at lung cancer diagnosis, *N* (%)	27 (63)
EGFR mutation, *N* (%)	7 (16)
Curative intended treatment, *N* (%)	6 (17)
Smoking, *N* (%)	
—Never	4 (9)
—Former	25 (58)
—Currently	12 (28)
—NA	2 (5)

Abbreviations: EGFR, epidermal growth factor receptor, NA, not available.

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
