# Peer review of "Neurofilament Light Chain as A Biomarker for Brain Metastases"

_cancers, 2020, doi:10.3390/cancers12102852_

Round 1

Reviewer 1 Report

In their paper, the authors present data that Neurofilament light chain may be an interesting serum biomarker for brain metastasis. They make their observations in a group of 68 lung cancer patients. The paper is very well written, easy to read, and the data are presented in a very clear way. The data are sound, and the analysis done in an appropriate way. Reading back the major issue that I wrote down reading the manuscript, it appeared that they mostly refer to the conclusions of the authors, including the possible relevance of the measurements for individual patients in the future.

Major comments:

In Figure 1 it is clear that – despite the significant p-values, there is a large overlap in absolute NfL values between the groups. Did the authors try to calculate the positive/negative predictive values? This could be as important as sensitivity and specificity. The question remains how this will affect individual patients. How can this be used in an N=1 situation.  

In the graphs (Fig 4) it can be seen (and this has been discussed by the authors) that also treatment may affect Neurofilament light chain levels. Also, in the discussion (L146-151) they indicate other additional factors that may influence NfL levels. So again, one has to ask how to interpret NfL levels in individual cases, e.g. how this biomarker could/should be implemented in the clinic.

A quick look at the literature indicates that increased levels of NfL are associated with a variety of clinical situation, some of them indicated in the discussion. A very recent paper (published Aug 2020 by Häring et al) states that longitudinal measurements of NfL would be much more informative than single measurements. It is understandable that this paper was not included in the present discussion. But it might be interesting to refer to it when updating the paper.

L100: It appears that longitudinal measurements are the way to go, starting at the day of diagnosis. Also to be able to correct for effects of treatment on NfL levels.

L127-129: It is a little unclear what has been done here. What do the authors mean by targeted therapy?? Looking at Figure 4, it appears that also persons with non-targetted therapy (like chemo and/or radiation therapy) should be excluded. On the other hand, when the authors say “brain metastasis at time of lung cancer diagnosis”, does this not in itself imply that the patients are completely treatment naïve??

Also: this appears to be an important analysis. Can the authors indicate the size of the two groups.

L172-173. I do not fully agree with this sentence. Reading the paper, I have the impression that not the absolute NfL level is indicative of a brain metastasis, but rather the longitudinal increase. In fact all figures in the paper indicate that baseline levels of NfL vary a lot, and are often higher than NfL levels observed in patients with brain metastasis.

Minor remarks.

The references may have to be renumbered, as in the current layout the Material and Methods are positioned after the discussion. Also, references 14-16 seem to be superfluous.

L159. The authors claim that an increase could be observed up to 5 months before symptoms occur. That is a rather optimistic interpretation of the data.  In most cases the situation very different.

Fig 4: in the legend the authors refer to a red dotted line. This line was not visible in the PDF file that I obtained. The Fig is also slightly misleading as the scale of the y-axis’s is different in the different graphs. E.g. compare 1728 with 1113.

Why are three out of nine graphs (Figure 4) presented as supplementary data? I would suggest to include all graphs in the manuscript.

For several graphs, the font and the lining could be a little thinner (Fig. 2 and 3). If needed,  Fig 2 and 3 could be combined in order to gain some space to also include the supplementary graphs in fig 4.

There appears to be a typo in the legend of the supplementary figure. Also the #-sign is not explained.

Reviewer 2 Report

  1. This is an interesting observation, and the idea of a surveillance biomarker for cerebral metastasis is an attractive one, it certainly would be much less burdensome than CT for patients and clinicians. However the numbers here overall are probably too small to draw any robust conclusions.
  2. It seems as though NFL is akin to a kind of "neuronal CRP" - and the authors have rightly identified that peripheral nerve damage in the context of treatment needs to be explored.  Given that it seems to be a general marker it would be interesting to know does the level correlate to cerebral metastatic burden, either by size of stage - as you know stage IV being subdivided into A and B- oligometastasis or not - with differing survival implications.
  3. Following that line of enquiry does NFL correlate to any other serum biomarkers or inflammation or injury? Did any of these patients receive steroids following diagnosis?
  4. The phenomenon of transient peaks following treatment is interesting. It would be interesting to compare NFL levels to a control group receiving treatment with radiological absence of metastasis. 

Round 2

Reviewer 1 Report

The authors have more than adequately responded to my comments. I have no further comments and/or questions.